# Test3R: Learning to Reconstruct 3D at Test Time

**Yuheng Yuan**[1]    **Qiuhong Shen**[1]    **Shizun Wang**[1]    **Xingyi Yang**[2,1]    **Xinchao Wang**[1]*

[1]National University of Singapore,  [2]The Hong Kong Polytechnic University

yuhengyuan@u.nus.edu, xingyi.yang@polyu.edu.hk, xinchao@nus.edu.sg

*Project page:* https://test3r-nop.github.io/

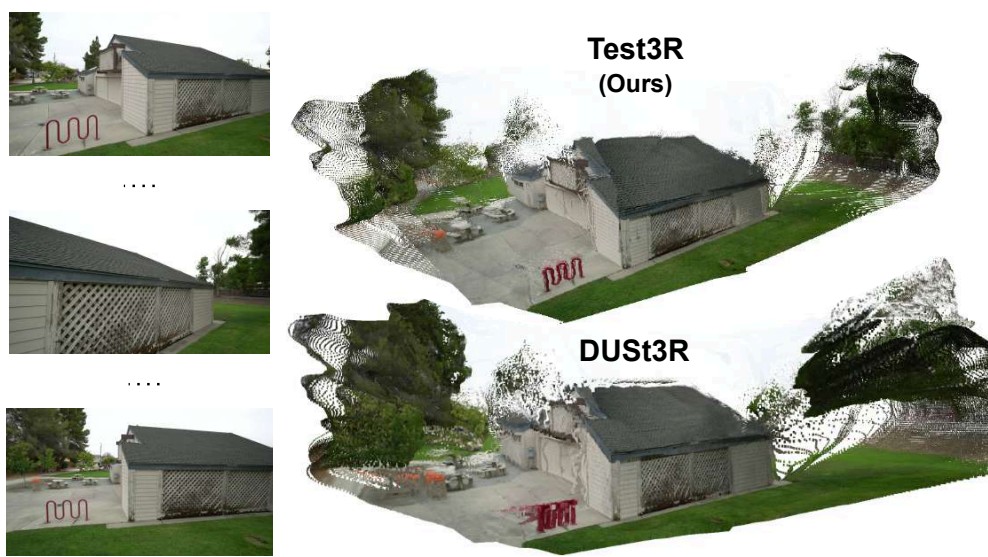

Figure 1: Given a set of images of a specific scene, **Test3R** improves the quality of reconstruction by maximizing the consistency between the pointmaps generated from multiple image pairs.

## Abstract

Dense matching methods like DUSt3R regress pairwise pointmaps for 3D reconstruction. However, the reliance on pairwise prediction and the limited generalization capability inherently restrict the global geometric consistency. In this work, we introduce **Test3R**, a surprisingly simple test-time learning technique that significantly boosts geometric accuracy. Using image triplets $(I_1, I_2, I_3)$, Test3R generates reconstructions from pairs $(I_1, I_2)$ and $(I_1, I_3)$. The core idea is to optimize the network at test time via a self-supervised objective: maximizing the geometric consistency between these two reconstructions relative to the common image $I_1$. This ensures the model produces cross-pair consistent outputs, regardless of the inputs. Extensive experiments demonstrate that our technique significantly outperforms previous state-of-the-art methods on the 3D reconstruction and multiview depth estimation tasks. Moreover, it is universally applicable and nearly cost-free, making it easily applied to other models and implemented with minimal test-time training overhead and parameter footprint. Code is available at https://github.com/nopQAQ/Test3R.

---

*Corresponding Author

39th Conference on Neural Information Processing Systems (NeurIPS 2025).

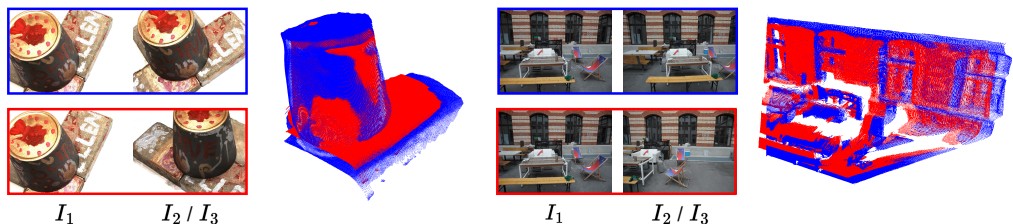

$I_1$      $I_2 / I_3$                           $I_1$      $I_2 / I_3$

Figure 2: **Inconsistency Study.** On the left are two image pairs sharing the same reference view $I_1$ but with different source views $I_2$ and $I_3$. On the right are the corresponding point maps, with each color indicating the respective image pair.

## 1 Introduction

3D reconstruction from multi-view images is a cornerstone task in computer vision. Traditionally, this process has been achieved by assembling classical techniques such as keypoint detection [1–3] and matching [4, 5], robust camera estimation [4, 6], Structure-from-Motion(SfM), Bundle Adjustment(BA) [7–9], and dense Multi-View Stereo [10, 11]. Although effective, these multi-stage methods require significant engineering effort to manage the entire process. This complexity inherently constrains their scalability and efficiency.

Recently, dense matching methods, such as DUSt3R [12] and MASt3R [13], have emerged as compelling alternatives. At its core, DUSt3R utilizes a deep neural network trained to *predict dense correspondences* between image pairs in an end-to-end fashion. Specifically, DUSt3R takes in two images and, for each, predicts a *pointmap*. Each pointmap represents the 3D coordinates of every pixel, as projected into a common reference view's coordinate system. Once pointmaps are generated from multiple views, DUSt3R aligns them by optimizing the registration of these 3D points. This process recovers the camera pose for each view and reconstructs the overall 3D geometry.

Despite its huge success, this *pair-wise prediction* paradigm is inherently problematic. Under such a design, the model considers only two images at a time. Such a constraint leads to several issues.

To investigate this, we compare the pointmaps of image $I_1$ but with different views $I_2$ and $I_3$ in Figure 2. It demonstrates that the predicted pointmaps are *imprecise* and *inconsistent*. Firstly, the precision of geometric predictions can suffer because the model is restricted to inferring scene geometry from just one image pair. This is especially true for *short-baseline* cases [14], where small camera movement leads to poor triangulation and thus inaccurate geometry. Second, reconstructing an entire scene requires pointmaps from multiple image pairs. Unfortunately, these individual pairwise predictions may not be mutually consistent. For example, the pointmap predicted from $(I_1, I_2, I_3)$ may not align with the prediction from $(I_1, I_2, I_3)$, as highlighted by the color difference in Figure 2. This local inconsistency further leads to discrepancies in the overall reconstruction. What makes things worse, the model, like many deep learning systems, struggles to generalize to new or diverse scenes. Such limitations directly exacerbate the previously discussed problems of precision and inter-pair consistency. Consequently, even with a final global refinement stage, inaccurate pointmaps lead to persistent errors.

To address these problems, in this paper, we present **Test3R**, a novel yet strikingly simple solution for 3D reconstruction, operating entirely *at test time*. Its core idea is straightforward: Maximizing the consistency between the reconstructions generated from multiple image pairs. This principle is realized through two basic steps:

1. Given image triplets $(I_1, I_2, I_3)$, Test3R first estimates two initial pointmaps with respect to $I_1$: $X_1$ from pairs $(I_1, I_2)$ and $X_2$ from $(I_1, I_3)$.

2. Test3R optimizes the network, so that the two pointmaps are cross-pair consistent, i.e., $X_1 \approx X_2$. Critically, this optimization is performed at test time via prompt tuning [15].

Despite its simplicity, Test3R offers a robust solution to all challenges mentioned above. It ensures consistency by aligning local two-view predictions, which resolves inconsistencies. This same mechanism also improves geometric precision: if a pointmap from short-baseline images is imprecise,

Test3R pushes it closer to an overall global prediction, which reduces errors. Finally, Test3R adapts to new, unseen scenes, minimizing its errors on unfamiliar data.

We evaluated Test3R on the DUSt3R for 3D reconstruction and multi-view depth estimation. Test3R performs exceptionally well across diverse datasets, improving upon vanilla DUSt3R to achieve competitive or state-of-the-art results in both tasks. Surprisingly, for multi-view depth estimation, Test3R even surpasses baselines requiring camera poses and intrinsics, as well as those trained on the same domain. This further validates our model's robustness and efficacy.

The best part is that Test3R is universally applicable and nearly cost-free for these pair-wise methods. This means it can easily be applied to other models sharing a similar pipeline. We validated this by incorporating our design into two categories of models, pairwise methods like MonST3R [16] and MASt3R [13], and multi-view models like VGGT [17]. Experimental results confirmed substantial performance improvements for both models.

The contributions of this work are as follows:

- We introduce **Test3R**, a novel yet simple solution to learn the reconstruction at test time. It optimizes the model via visual prompts to maximize the cross-pair consistency. It provides a robust solution to the challenges of the pairwise prediction paradigm and limited generalization capability.
- We conducted comprehensive experiments across several downstream tasks on the DUSt3R. Experiment results demonstrate that Test3R not only improves the reconstruction performance compared to vanilla DUSt3R but also outperforms a wide range of baselines.
- Our design is universally applicable and nearly cost-free. It can be easily applied to other models and implemented with minimal test-time training overhead and parameter footprint.

## 2 Related Work

### 2.1 Multi-view Stereo

Multi-view Stereo(MVS) aims to densely reconstruct the geometry of a scene from multiple overlapping images. Traditionally, all camera parameters are often estimated with SfM [18], as the given input. Existing MVS approaches can generally be classified into three categories: traditional hand-crafted [11, 19–21], global optimization [22–25], and learning-based methods [10, 26–29]. Recently, DUSt3R [12] has attracted significant attention as a representative of learning-based methods. It attempts to estimate dense pointmaps from a pair of views without any explicit knowledge of the camera parameters. Subsequent tremendous works focus on improving its efficiency [17, 30, 31], quality [13, 17, 32], and broadening its applicability to dynamic reconstruction [16, 33–35] and 3D perception [36]. The majority employ the pairwise prediction strategy introduced by DUSt3R [12]. However, the pair-wise prediction paradigm is inherently problematic. It leads to low precision and mutually inconsistent pointmaps. Furthermore, the limited generalization capability of the model exacerbates these issues. This challenge continues even with the latest models [17, 37], which can process multiple images in a single forward pass. While potentially more robust, these newer approaches demand significantly larger resources for training and, importantly, still face challenges in generalizing to unseen environments. To this end, we introduce a novel test-time training technique. This simple design ensures the cross-pairs consistency by aligning local two-view predictions to push the pointmaps closer to an overall global prediction, which addresses all challenges mentioned above.

### 2.2 Test-time Training

The idea of training on unlabeled test data dates back to the 1990s [38], called transductive learning. As Vladimir Vapnik [39] famously stated, "Try to get the answer that you really need but not a more general one", this principle has been widely applied to SVMs [40, 41] and recently in large language models [42]. Another early line of work is local learning [43, 44]: for each test input, a "local" model is trained on the nearest neighbors before a prediction is made. Recently, Test-time training(TTT) [45] proposes a general framework for test-time training with self-supervised learning, which produces a different model for every single test input through the self-supervision task. This strategy allows the model trained on the large-scale datasets to adapt to the target domain at test time. Many other works have followed this framework since then [46–49]. Inspired by these studies, we introduce Test3R, a novel yet simple technique that extends the test-time training paradigm to the 3D reconstruction

domain. Our model exploits the cross-pairs consistency as a strong self-supervised objective to optimize the model parameters at test time, thereby improving the final quality of reconstruction.

## 2.3 Prompt tuning

Prompt tuning was first proposed as a technique that appends learnable textual prompts to the input sequence, allowing pre-trained language models to adapt to downstream tasks without modifying the backbone parameters [50]. In follow-up research, a portion of studies [51, 52] explored strategies for crafting more effective prompt texts, whereas others [53–55] proposed treating prompts as learnable, task-specific continuous embeddings, which are optimized via gradient descent during fine-tuning referred to as Prompt Tuning. In recent years, prompt tuning has also received considerable attention in the 2D vision domain. Among these, Visual Prompt Tuning (VPT) [15] has gained significant attention as an efficient approach specifically tailored for vision tasks. It introduces a set of learnable prompt tokens into the pretrained model and optimizes them using the downstream task's supervision while keeping the backbone frozen. This strategy enables the model to transfer effectively to downstream tasks. In our study, we leverage the efficient fine-tuning capability of VPT to optimize the model to ensure the pointmaps are cross-view consistent. This design makes our model nearly cost-free, requiring minimal test-time training overhead and a small parameter footprint.

## 3    Preliminary of DUSt3R

Given a set of images $\{\mathbf{I}_k\}$ of a specific scene, DUSt3R [12] achieves high precision 3D reconstruction by predicting pairwise pointmaps of all views and global alignment.

**Pairwise prediction.** Briefly, DUSt3R takes a pair of images, $I^1, I^2 \in \mathbb{R}^{W \times H \times 3}$ as input and outputs the corresponding pointmaps $X^{1,1}, X^{2,1} \in \mathbb{R}^{W \times H \times 3}$ which are expressed in the same coordinate frame of $I^1$. In our paper, we refer to the viewpoint of $I^1$ as the reference view, while the other is the source view. Therefore, the pointmaps $X^{1,1}, X^{2,1}$ can be denoted as $X^{ref,ref}, X^{src,ref}$, respectively.

In more detail, these two input images $I^{ref}, I^{src}$ are first encoded by the same weight-sharing ViT-based model [56] with $N_e$ layers to yield two token representations $F^{ref}$ and $F^{src}$:

$$F^{ref} = Encoder(I^{ref}), \quad F^{src} = Encoder(I^{src}) \tag{1}$$

After encoding, the network reasons over both of them jointly in the decoder. Each decoder block also attends to tokens from the other branch:

$$G_i^{ref} = DecoderBlock_i^{ref}(G_{i-1}^{ref}, G_{i-1}^{src}) \tag{2}$$

$$G_i^{src} = DecoderBlock_i^{src}(G_{i-1}^{src}, G_{i-1}^{ref}) \tag{3}$$

where $i = 1, \cdots, N_d$ for a decoder with $N_d$ decoder layers and initialized with encoder tokens $G_0^{ref} = F^{ref}$ and $G_0^{src} = F^{src}$. Finally, in each branch, a separate regression head takes the set of decoder tokens and outputs a pointmap and an associated confidence map:

$$X^{ref,ref}, C^{ref,ref} = Head^{ref}(G_0^{ref}, \ldots, G_{N_d}^{ref}), \tag{4}$$

$$X^{src,ref}, C^{src,ref} = Head^{src}(G_0^{src}, \ldots, G_{N_d}^{src}). \tag{5}$$

**Global alignment.** After predicting all the pairwise pointmaps, DUSt3R introduces a global alignment to handle pointmaps predicted from multiple images. For the given image set $\{\mathbf{I}_t^i\}_{i=1}^{N_t}$, DUSt3R first constructs a connectivity graph $\mathcal{G}(\mathcal{V}, \mathcal{E})$ for selecting pairwise images, where the vertices $\mathcal{V}$ represent $N_t$ images and each edge $e \in \mathcal{E}$ is an image pair. Then, it estimates the depth maps $D := \{\mathbf{D}_k\}$ and camera pose $\pi := \{\pi_k\}$ by

$$\underset{\mathbf{D}, \pi, \sigma}{\arg\min} \sum_{e \in \mathcal{E}} \sum_{v \in e} \mathbf{C}_v^e \|\mathbf{D}_v - \sigma_e P_e(\pi_v, \mathbf{X}_v^e)\|_2^2, \tag{6}$$

where $\sigma = \{\sigma_e\}$ are the scale factors defined on the edges, $P_e(\pi_v, \mathbf{X}_v^e)$ means projecting the predicted pointmap $\mathbf{X}_v^e$ to view $v$ using poses $\pi_v$ to get a depth map. The objective function in eq. (6) explicitly constrains the geometry alignment between frame pairs, aiming to preserve cross-view consistency in the depth maps.

# 4  Methods

Test3R is a test-time training technique that adapts DUSt3R [12] to challenging test scenes. It improves reconstruction by maximizing cross-pair consistency. We begin by analyzing the root cause of inconsistency in Sec. 4.1. In Sec. 4.2, we establish the core problem and define the test-time training objective. Finally, we employ prompt tuning for efficient test-time adaptation in Sec. 4.3.

## 4.1  Cross-pair Inconsistency

DUSt3R [12] aims to achieve consistency through global alignment; however, the inaccurate and inconsistent pointmaps lead to persistent errors, significantly compromising the effectiveness of global alignment.

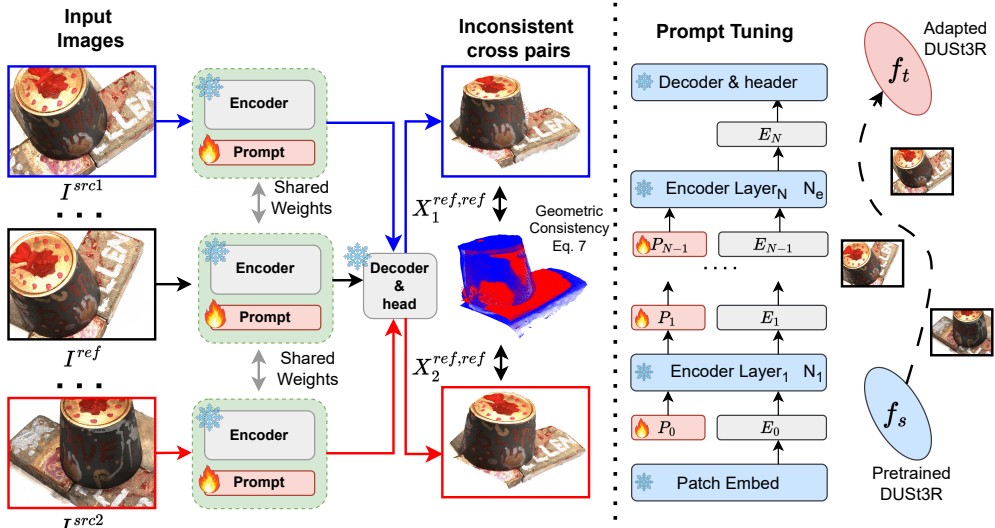

Figure 3: **Overview of Test3R.** The primary goal of Test3R is to adapt a pretrained reconstruction model $f_s$ to the specific distribution of test scenes $f_t$. It achieves this goal by optimizing a set of visual prompts at test time through a self-supervised training objective that maximizes cross-pair consistency between $X_1^{ref,ref}$ and $X_2^{ref,ref}$.

Therefore, we show a qualitative analysis of the quality of pointmaps on the DTU [57] and ETH3D [58] datasets. Specifically, we compare the pointmap for the same reference view but paired with two different source views, and align these two pointmaps to the same coordinate system using Iterative Closest Point (ICP). The result is shown in Figure 2. On the left are two image pairs sharing the same reference view but with different source views. On the right are the corresponding pointmaps, with each color indicating the respective image pair.

Observations. These two predicted pointmaps of the reference view exhibit inconsistencies, as highlighted by the presence of large regions with inconsistent colors in 3D space. Ideally, if these pointmaps are consistent, they should be accurate enough to align perfectly in 3D space, resulting in a single, unified color (either blue or red). This result indicates that DUSt3R may produce different pointmaps for the same reference view when paired with different source views.

In our view, this phenomenon stems from the problematic pair-wise prediction paradigm. First, since only two views are provided as input at each prediction step, the scene geometry is estimated solely based on visual correspondences between a single image pair. Therefore, the model produces inaccurate pointmaps. Second, all predicted pointmaps are mutually inconsistent individual pairs. For different image pairs, their visual correspondences are also different. As a result, DUSt3R may produce inconsistent pointmaps for the same reference view when paired with different source views due to the different correspondences. This issue significantly hinders the effectiveness of subsequent global alignment and further leads to discrepancies in the overall reconstruction. What's worse, the limited generalization capability of DUSt3R further exacerbates the above issues of low precision and cross-pair inconsistency.

## 4.2 Triplet Objective Made Consistent

The inconsistencies observed above highlight a core limitation of the pairwise prediction paradigm. Specifically, DUSt3R may produce different pointmaps for the same reference view when paired with different source views. This motivates a simple but effective idea: enforce triplet consistency across these pointmaps directly at test time, as shown in Figure 3.

**Definition.** We first describe the definition of test-time training on the 3D reconstruction task, where only images $\{I_t^i\}_{i=1}^{N_t}$ from the test scene are available. During training time training phase, $N_s$ labeled samples $\{I_s^i, \bar{X}_s^i\}_{i=1}^{N_s}$ collected from various scenes are given, where $I_s^i \in \mathcal{I}_s$ and $\bar{X}_s^i \in \bar{\mathcal{X}}_s$ are images and the corresponding pointmaps derived from the ground-truth depth $\bar{D}_s \in \bar{\mathcal{D}}_s$. Furthermore, we denote DUSt3R [12], parameterized by $\theta$, as the model trained to learn the reconstruction function $f_s : \mathcal{I}_s \to \bar{\mathcal{X}}_s$. Subsequently, during test time training phase, only unlabeled images $\{I_t^i\}_{i=1}^{N_t}$ from test scene are available, where $I_t^i \in \mathcal{I}_t$. Our goal is to optimize the model $f_s$ to the specific scene $f_t : \mathcal{I}_t \to \bar{\mathcal{X}}_t$ at test time. This is achieved by minimizing the self-supervised training objective $\ell$.

Specifically, our core training objective is to maximize the geometric consistency by aligning the pointmaps of the reference view when paired with different source views. For a set of images $\{I_t^i\}_{i=1}^{N_t}$ from the specific scene, we consider a triplet consisting of one reference view and two different source views, denoted as $(I^{ref}, I^{src1}, I^{src2})$. Subsequently, Test3R forms two reference–source view pairs $(I^{ref}, I^{src1})$ and $(I^{ref}, I^{src2})$ from this triplets. These reference–source view pairs are then fed into the Test3R independently to predict pointmaps of reference views under different source view conditions in the same coordinate frame of $I^{ref}$, denoted as $\mathbf{X}_1^{ref,ref}$ and $\mathbf{X}_2^{ref,ref}$. Finally, we construct the training objective by aligning these two inconsistent pointmaps, formulated as:

$$\ell = \left\| X_1^{ref,ref} - X_2^{ref,ref} \right\|_1. \tag{7}$$

With this objective, we can collectively compose triplets from a large number of views of an unseen 3D scene at test time. It guides the model to successfully resolve the limitations mentioned in Section 4.1. For inconsistencies, it ensures consistency by aligning the local two-view predictions. Meanwhile, it also pushes the predicted pointmap closer to an overall global prediction to mitigate the inaccuracy. Moreover, by optimizing for the specific scene at test time, it enables the model to adapt to the distribution of that scene.

## 4.3 Visual Prompt Tuning for Test Time Training

After the self-supervised training objective is defined, effectively modulating the model during test-time training for specific scenes remains a non-trivial challenge. During the test-time training phase, it only relies on unsupervised training objectives. However, these objectives are often noisy and unreliable, which makes the model prone to overfitting and may lead to training collapse, especially when only a limited number of images are available for the current scene. Fortunately, similar issues has been partially explored in the 2D vision community. In these works, visual prompt tuning [15] has demonstrated strong effectiveness in domain adaptation in 2D classification tasks [59]. It utilizes a set of learnable continuous parameters to learn the specific knowledge while retaining the knowledge learned from large-scale pretraining. Motivated by this, we explore the use of visual prompts as a carrier to learn the geometric consistency for specific scenes.

Specifically, we incorporate a set of learnable prompts into the encoder of DUSt3R [12]. Consider an encoder of DUSt3R with $N_e$ standard Vision Transformer(ViT) [56] layers, an input image is first divided into fixed-sized patches and then embedded into d-dimensional tokens $\mathbf{E_0} = \{\mathbf{e}_0^k \in \mathbb{R}^D | k \in \mathbb{N}, 1 \le k \le N_t\}$, where $N_t$ is the length of image patch tokens. Subsequently, to optimize the model, we introduce a set of learnable prompt tokens $\{\mathbf{P}_{i-1}\}_{i=1}^{N_e}$ into each Transformer layer. For $i-th$ transformer layer, the prompt tokens are denoted as $\mathbf{P}_{i-1} = \{\mathbf{p}_{i-1}^k \in \mathbb{R}^D | k \in \mathbb{N}, 1 \le k \le N_p\}$, where $N_p$ is the length of prompt tokens. Therefore, the encoder layer augmented by visual prompts is formulated as:

$$[\_, \mathbf{E_i}] = L_i([\mathbf{P_{i-1}}, \mathbf{E_{i-1}}]) \tag{8}$$

where $\mathbf{P}_{i-1}$ and $\mathbf{E}_{i-1}$ are learnable prompt tokens and image patch tokens at $i-1$-th Transformer layer.

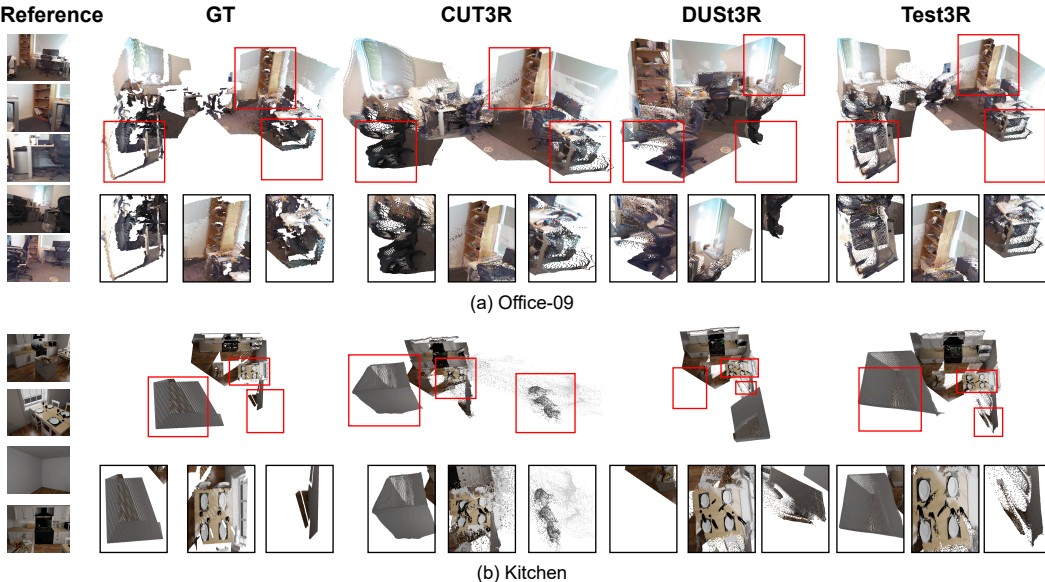

| Reference | GT | CUT3R | DUSt3R | Test3R |

(a) Office-09

(b) Kitchen

Figure 4: **Qualitative Comparison on 3D Reconstruction.**

**Test-time training.** We only fine-tune the parameters of the prompts, while all other parameters are fixed. This strategy enables our model to maximize geometric consistency by optimizing the prompts at test time while retaining the reconstruction knowledge acquired from large-scale datasets training within the unchanged backbone.

# 5 Experiment

We evaluate our method across a range of 3D tasks, including 3D Reconstruction( Section 5.1) and Multi-view Depth( Section 5.2). Moreover, we discuss the generalization of Test3R and the prompt design( Section 5.3). Additional experiments and detailed information are provided in the appendix.

**Baselines.** Our primary baseline is DUSt3R [12], which serves as the backbone of our technique in the experiment. Subsequently, we select different baselines for the specific tasks to comprehensively evaluate the performance of our proposed method. For the 3D reconstruction task, which is the primary focus of the majority of 3R-series models, we compared our method with current mainstream approaches to evaluate its effectiveness. It includes MAST3R [13], MonST3R [16], CUT3R [35] and Spann3R [31]. All of these models are follow-up works building on the foundation established by DUSt3R [12]. Furthermore, for the multi-view depth estimation task, we not only compare our model with baselines [60, 61] that do not require camera parameters but also evaluate our model against methods [9, 11, 28, 60–63, 63, 64] that rely on camera parameters or trained on datasets from the same distribution to demonstrate the effectiveness of our technique.

## 5.1 3D Reconstruction

We utilize two scene-level datasets, 7Scenes [65] and NRGBD [66] datasets. We follow the experiment setting on the CUT3R [35], and employ several commonly used metrics: Accuracy (Acc), Completion (Comp), and Normal Consistency (NC) metrics. Each scene has only 3 to 5 views available for the 7Scenes [65] dataset and 2 to 4 views for NRGBD [66] dataset. This is a highly challenging experimental setup, as the overlap between images in each scene is minimal, demanding a strong scene reconstruction capability.

**Quantitative Results.** The quantitative evaluation is shown in Table 1. Compared to vanilla DUSt3R [12], our model demonstrates superior performance, outperforming DUSt3R on the majority of evaluation metrics, particularly in terms of mean accuracy and completion. Moreover, our approach achieves comparable or even superior results compared to mainstream methods. Only CUT3R [35] and MAST3R [13] outperform our approach on several metrics. This demonstrates the effectiveness of our test-time training strategy.

**Qualitative Results.** The qualitative results are shown in Figure 4. We compare our method with CUT3R [35] and DUSt3R [12] on the *Office* and *Kitchen* scenes from the 7Scenes [65] and NRGBD [66] datasets, respectively. We observe that DUSt3R incorrectly regresses the positions of scene views, leading to errors in the final scene reconstruction. In contrast, our model achieves more reliable scene reconstructions. This improvement is particularly evident in the statue in the *Office* scene and the wall in the *Kitchen* scene. For these two objects, the reconstruction results from DUSt3R are drastically different from the ground truth. Compared to CUT3R [35], the current state-of-the-art in 3D reconstruction, we achieve better reconstruction results. Specifically, we effectively avoid the generation of outliers, resulting in more accurate pointmaps. Details can be seen in the red bounding boxes as shown in Figure 4.

Table 1: **3D reconstruction comparison on 7Scenes and NRGBD datasets.**

| | 7Scenes | | | | | | NRGBD | | | | | |
| | Acc↓ | | Comp↓ | | NC↑ | | Acc↓ | | Comp↓ | | NC↑ | |
| Method | Mean | Med. | Mean | Med. | Mean | Med. | Mean | Med. | Mean | Med. | Mean | Med. |
|---|---|---|---|---|---|---|---|---|---|---|---|---|
| MAST3R [13] | 0.189 | 0.109 | 0.211 | 0.110 | 0.687 | 0.766 | 0.085 | 0.033 | 0.063 | 0.028 | 0.794 | 0.928 |
| MonST3R [16] | 0.240 | 0.180 | 0.268 | 0.167 | 0.672 | 0.758 | 0.272 | 0.114 | 0.287 | 0.110 | 0.758 | 0.843 |
| Spann3R [31] | 0.298 | 0.226 | 0.205 | 0.112 | 0.650 | 0.730 | 0.416 | 0.323 | 0.417 | 0.285 | 0.684 | 0.789 |
| CUT3R [35] | 0.126 | 0.047 | 0.154 | 0.031 | 0.727 | 0.834 | 0.099 | 0.031 | 0.076 | 0.026 | 0.837 | 0.971 |
| DUSt3R [12] | 0.146 | 0.078 | 0.181 | 0.067 | 0.736 | 0.839 | 0.144 | 0.019 | 0.154 | 0.018 | 0.871 | 0.982 |
| **Test3R(Ours)** | 0.105 | 0.051 | 0.136 | 0.035 | 0.746 | 0.855 | 0.083 | 0.021 | 0.079 | 0.019 | 0.870 | 0.983 |

Table 2: **Multi-view depth evaluation.** (Parentheses) denote training on data from the same domain.

| Method | GT Pose | GT Range | GT Intrinsics | Align | DTU rel↓ | DTU τ↑ | ETH3D rel↓ | ETH3D τ↑ | AVG rel↓ | AVG τ↑ |
|---|---|---|---|---|---|---|---|---|---|---|
| COLMAP [9, 11] | ✓ | ✗ | ✓ | ✗ | 0.7 | 96.5 | 16.4 | 55.1 | 8.6 | 75.8 |
| COLMAP Dense [9, 11] | ✓ | ✗ | ✓ | ✗ | 20.8 | 69.3 | 89.8 | 23.2 | 55.3 | 46.3 |
| MVSNet [28] | ✓ | ✓ | ✓ | ✗ | (1.8) | (86.0) | 35.4 | 31.4 | 18.6 | 58.7 |
| Vis-MVSSNet [63] | ✓ | ✓ | ✓ | ✗ | (1.8) | (87.4) | 10.8 | 43.3 | 6.3 | 65.4 |
| MVS2D ScanNet [64] | ✓ | ✓ | ✓ | ✗ | 17.2 | 9.8 | 27.4 | 4.8 | 22.3 | 7.3 |
| MVS2D DTU [64] | ✓ | ✓ | ✓ | ✗ | (3.6) | (64.2) | 99.0 | 11.6 | 51.3 | 37.9 |
| DeMoN [60] | ✓ | ✗ | ✓ | ✗ | 23.7 | 11.5 | 19.0 | 16.2 | 21.4 | 13.9 |
| DeepV2D KITTI [61] | ✓ | ✗ | ✓ | ✗ | 24.6 | 8.2 | 30.1 | 9.4 | 27.4 | 8.8 |
| DeepV2D ScanNet [61] | ✓ | ✗ | ✓ | ✗ | 9.2 | 27.4 | 18.7 | 28.7 | 14.0 | 28.1 |
| MVS2D ScanNet [64] | ✓ | ✗ | ✓ | ✗ | 5.0 | 57.9 | 30.7 | 14.4 | 17.9 | 36.2 |
| Robust MVD Baseline [62] | ✓ | ✗ | ✓ | ✗ | 2.7 | 82.0 | 9.0 | 42.6 | 5.9 | 62.3 |
| DeMoN [60] | ✗ | ✗ | ✓ | $\|\|t\|\|$ | 21.8 | 16.6 | 17.4 | 15.4 | 19.6 | 16.0 |
| DeepV2D KITTI [61] | ✗ | ✗ | ✓ | med | 24.8 | 8.1 | 27.1 | 10.1 | 26.0 | 9.1 |
| DeepV2D ScanNet [61] | ✗ | ✗ | ✓ | med | 7.7 | 33.0 | 11.8 | 29.3 | 9.8 | 62.3 |
| DUSt3R [1] | ✗ | ✗ | ✗ | med | 3.3 | 69.9 | 3.3 | 73.0 | 3.3 | 71.5 |
| **Test3R(Ours)** | ✗ | ✗ | ✗ | med | 2.0 | 84.1 | 3.2 | 74.0 | 2.6 | 79.1 |

## 5.2 Multi-view Depth

Following RobustMVD [62], performances are measured on the object-centric dataset DTU [57] and scene-centric dataset ETH3D [58]. To evaluate the depth map, we report the Absolute Relative Error (rel) and the Inlier Ratio ($\tau$) at a threshold of 3% on each test set and the averages across all test sets.

**Quantitative Results.** The quantitative evaluation is shown in Table 2. On the DTU dataset, our model significantly improves upon the performance of vanilla DUSt3R, reducing the Absolute Relative Error by 1.3 and increasing the Inlier Ratio by 14.2. Similarly, on the ETH3D dataset, our model also demonstrates comparable improvements, achieving state-of-the-art performance on this challenging benchmark as well. Notably, our model surpasses the majority of methods that rely on camera poses and intrinsic parameters, and the models trained on the dataset from the same domain. This indicates that our approach effectively captures scene-specific global information and enables the adaptation of the distribution of test scenes, thereby significantly improving the quality of the depth maps.

**Qualitative Results.** The qualitative result is shown in Figure 5. We present the depth map on the key view, following RobustMVD [62]. We observe that Test3R effectively improves the accuracy of depth estimation compared to DUSt3R and RobustMVD [62] with camera parameters. Specifically, Test3R captures more fine-grained details, including the computer chassis and table. Additionally, on the white-background DTU dataset, Test3R effectively understands scene context, allowing it to accurately estimate the depth of background regions.

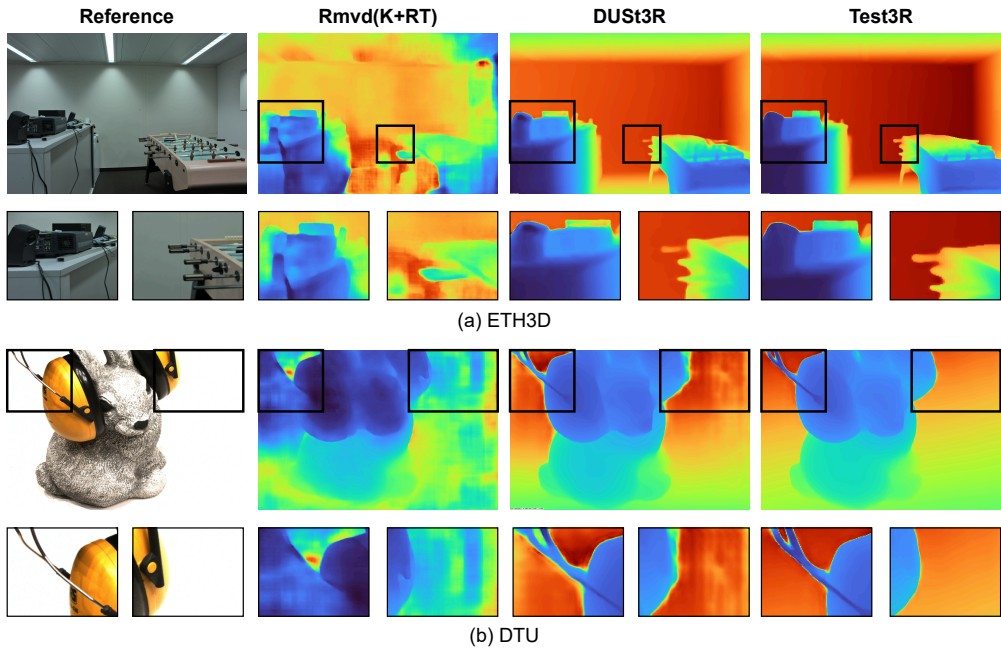

| Reference | Rmvd(K+RT) | DUSt3R | Test3R |

(a) ETH3D

(b) DTU

Figure 5: **Qualitative Comparison on Multi-view Depth.**

## 5.3 Ablation Study and Analysis

### 5.3.1 Framework Generalization.

To demonstrate the generalization ability of our proposed technique, we applied Test3R to MAST3R [13] and MonST3R [16], and evaluated the performances on the 7Scenes [65] dataset. As shown in Table 3, Test3R effectively improves the performance of MAST3R and MonST3R on 3D reconstruction task. This demonstrates the generalization ability of our technique, which can be applied to other models sharing a similar pipeline.

### 5.3.2 Ablation on Visual Prompt.

We introduce a model variant, Test3R-S, and conduct an ablation study to evaluate the impact of visual prompts. For Test3R-S, the prompts are only inserted into the first Transformer layer, accompany the image tokens through the encoding process, and are then discarded.

Table 3: **Generalization Study.**

| | 7Scenes | | | | | |
| | Acc↓ | | Comp↓ | | NC↑ | |
| Method | Mean | Med. | Mean | Med. | Mean | Med. |
|---|---|---|---|---|---|---|
| MAST3R [13] | 0.189 | 0.109 | 0.211 | 0.110 | 0.687 | 0.766 |
| MAST3R(w. Test3R) | 0.179 | 0.108 | 0.177 | 0.059 | 0.702 | 0.788 |
| MonST3R [16] | 0.240 | 0.180 | 0.268 | 0.167 | 0.672 | 0.758 |
| MonST3R(w. Test3R) | 0.218 | 0.167 | 0.251 | 0.160 | 0.687 | 0.775 |

Table 4: **Ablation study on Visual Prompt.**

| | Prompts Length | | | | | | | |
| Varients | 8 | | 16 | | 32 | | 64 | |
| | Acc↓ | Comp↓ | Acc | Comp↓ | Acc↓ | Comp↓ | Acc↓ | Comp↓ |
|---|---|---|---|---|---|---|---|---|
| Test3R-S | 0.133 | 0.142 | 0.125 | 0.159 | 0.120 | 0.158 | 0.119 | 0.163 |
| Test3R | 0.118 | 0.131 | 0.122 | 0.155 | 0.105 | 0.136 | 0.149 | 0.170 |

The result is shown in Table 4. Both Test3R-S and Test3R effectively improve model performance, compared to vanilla DUSt3R. For prompt length, we observe that when the number of prompts is small, increasing the prompt length can enhance the ability of Test3R to improve reconstruction quality. However, as the prompt length increases, the number of trainable parameters also grows, making it more challenging to converge within the same number of iterations, thereby reducing their overall effectiveness. For prompt insertion depth, we observe that Test3R, which uses distinct prompts at each layer, demonstrates superior performance. This is because the feature distributions vary across each layer of the encoder of DUSt3R, making layer-specific prompts more effective for fine-tuning. However, as the number of prompt parameters increases, Test3R becomes more susceptible to optimization challenges compared to Test3R-S, leading to a faster performance decline.

# 6   Conclusion

In this paper, we present Test3R, a novel yet strikingly simple solution that learns to reconstruct at test time. It maximizes the cross-pair consistency via optimizing a set of visual prompts at test time. This design successfully mitigates the reconstruction quality degradation caused by the pairwise predictions paradigm and limited generalization capability. Extensive experiments show that our simple design not only effectively improves model performance but also achieves state-of-the-art performance across various tasks. Moreover, our technique is universally applicable and nearly cost-free, which can be widely applied to different models and implemented with minimal test-time training overhead and parameter footprint.

## Acknowledgement

This project is supported by the National Research Foundation, Singapore, under its Medium Sized Center for Advanced Robotics Technology Innovation.

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
