# OpenReview forum: "Test3R: Learning to Reconstruct 3D at Test Time"
_NeurIPS.cc/2025/Conference — NeurIPS 2025 poster_

### Official Review · Reviewer_aSNX · 2025-07-01

**Clarity:** 3
**Significance:** 2
**Originality:** 2
**Rating:** 4
**Confidence:** 4

**Summary:**

This paper proposes Test3R, which improves 3D reconstruction accuracy through test-time training. Specifically, it optimizes consistency across image pairs using a triplet consistency objective, enhancing overall reconstruction quality. The method employs prompt tuning, adjusting only the prompt parameters, resulting in relatively low computational cost. Test3R surpasses some previous methods across several challenging datasets and tasks without relying on camera parameters. Test3R demonstrates strong generalization capabilities and can be integrated into various reconstruction frameworks based on DUSt3R.

**Questions:**

All my concerns have been detailed in the weaknesses section.

**Ethical Concerns:**

["NO or VERY MINOR ethics concerns only"]

**Final Justification:**

Thanks for the authors’ efforts in the rebuttal. I am impressed by the significant performance improvements achieved when integrating with VGGT, as well as the remarkably short optimization time of only 14 seconds. I decide to raise my score to borderline accept. And I would recommend the authors to incorporate some of key points mentioned in rebuttal into the the paper.

**Limitations:**

yes

**Quality:**

2

**Strengths And Weaknesses:**

# Strengths

1. The paper is clearly written and easy to follow.

2. The technical design of the proposed method is reasonable.

# Weaknesses

1. Although the triplet consistency-based optimization objective is technically sound, it is not an optimal approach since it does not fully exploit the information available in test scenes. A more effective approach might directly leverage images for optimization, as demonstrated by methods like 3DGS.

2. Recent feed-forward approaches such as VGGT can handle reconstruction with multiple input views without needing pair-wise depth estimation, theoretically avoiding the inconsistency issues discussed regarding DUSt3R. However, the authors lack discussion and comparison with such methods.

3. The proposed method requires per-scene optimization, which, even when tuning only part of the parameters, inevitably reduces speed. The paper seems to lack a thorough analysis and comparison of reconstruction speed, which is critical. If the method is slow, it would be beneficial for the authors to include comparisons with other per-scene optimization methods such as some 3DGS based methods.

4. Overall, the proposed approach is relatively simple, essentially adding per-scene optimization as a post-processing step to DUSt3R. Moreover, the chosen optimization objective is straightforward but clearly suboptimal, and the Visual Prompt Tuning component essentially utilizes a mature fine-tuning strategy. Thus, I consider the novelty and technical contribution of this paper to be very limited.

5. There are minor issues, such as the typo "recomstruction" on line 39.

---

> ### Author Rebuttal · Authors · 2025-07-31
>
> We thank the reviewer for the constructive comments. Below, we provide a point-by-point response to address your concerns. We welcome further discussion to improve the clarity and effectiveness of our work.
> > **`>>> W1:`**
> > **Not an optimal approach. A more effective approach might directly leverage images for optimization, as demonstrated by methods like 3DGS.**
> ---
>
> **`>>> A1:`**
> * **3DGS-based model.** We would like to clarify that our method and 3D Gaussian Splatting (3DGS) serve different purposes and are based on fundamentally different design principles. 3DGS is designed for photorealistic appearance reconstruction through iterative photometric optimization. In contrast, our approach focuses on accurate geometric prediction using feed-forward models, similar to DUSt3R [1], MAST3R [2], and VGGT [3]. These methods play a role similar to Structure-from-Motion (SfM) in the reconstruction pipeline. Therefore, direct comparisons with methods like 3DGS are not fully aligned in scope. Importantly, our method can provide accurate geometric initialization and camera poses, which can serve as a strong complement for iterative 3D reconstruction models such as 3DGS and NeRF.
>
> * **NOT suboptimal**. Our method is designed to achieve optimal geometric prediction under the feed-forward paradigm. By enforcing geometric consistency across triplets, our method leverages the mutual constraint between partial reconstructions, which is a fundamental signal in scene geometry. In an ideal scenario with perfect reconstruction, this consistency loss would naturally vanish. This principled design allows our model to generalize across scenes with minimal testing-time optimization, making it efficient and robust in practice.
>
> [1] DUSt3R: Geometric 3D Vision Made Easy.
>
> [2] Grounding Image Matching in 3D with MASt3R.
>
> [3] VGGT: Visual Geometry Grounded Transformer.
>
> > **`>>> W2:`**
> > **Discussion and comparison with recent feed-forward approaches such as VGGT.**
> ---
>
> **`>>> A2:`**
>
> * **Comparison**: We have provided a comparison with two feed-forward approaches, VGGT[1] and Fast3R[2], in the Tab.6 in the Appendices. As shown in this table, Test3R achieves significantly superior reconstruction quality compared to Fast3R. Moreover, it outperforms VGGT across the vast majority of evaluation metrics.
>
> * **Discussion**: While VGGT processes multiple views jointly and avoids pairwise depth estimation, it still suffers from cross-pair inconsistency in predicted geometries. We would like to highlight that this inconsistency can still be leveraged through our method to enhance the reconstruction quality of VGGT. To demonstrate this, we implemented our method on VGGT. Specifically, we insert the prompts into the image tokens of each input image, like the operation of VGGT’s handling of the register token. During test-time training, it still takes a triplet as input to predict two pointmaps under the same reference view but paired with different source views as introduced in Sec.4.2. We evaluated it on the Office-07 scene on the 7Scenes dataset. The final results are shown in the table below. It indicates that Test3R significantly enhances both Acc. and Comp. of pointmap predicted by VGGT[1].
>
> |  Method   | Acc&nbsp;$\downarrow$ | Comp&nbsp;$\downarrow$ | NC&nbsp;$\uparrow$ |
> | :-------: | :-------------------: | :--------------------: | :----------------: |
> |   VGGT    |         0.27          |          0.20          |        0.71        |
> | VGGT+Ours |         0.11          |          0.12          |        0.71        |
>
> [1] VGGT: Visual Geometry Grounded Transformer.
>
> [2] Fast3R: Towards 3D Reconstruction of 1000+ Images in One Forward Pass
>
> > **`>>> W3:`**
> > **A thorough analysis and comparison of reconstruction speed. Comparisons with other per-scene optimization methods, such as some 3DGS-based methods.**
> ---
>
> **`>>> A3:`**
> * **Reconstruction speed.** We appreciate the reviewer’s concern about runtime. The optimization process is relatively short, taking approximately 14 seconds on a single GPU. The majority of Test3R’s time overhead comes from the pair-wise prediction and global alignment in the original DUSt3R's pipeline, which together take 320 seconds for a set of 50 images on the 7Scenes dataset.
>
> * **Runtime comparison with 3DGS.** While these two methods are not directly comparable as discussed in W1,  our method only introduces a few seconds of extra overhead; the total reconstruction time remains much shorter than that required by 3DGS to reconstruct a large-scale scene, which typically takes about an hour.
>
>   We will include this discussion and timing statistics in the revision for clarity and completeness.
>
> > **`>>> W4:`**
> > **The limited novelty and technical contribution of this paper.**
> ---
> **`>>> A4:`** We thank the reviewer for this question. It gives us a valuable opportunity to clarify our design.
>
> Yes, our method is simple, but this simplicity is intentional and highlights our strength. Simplicity does not diminish novelty; rather, it underscores the effectiveness of our insight.
>
> Specifically, Test3R is novel in both **new paradim** and **technical application**:
>
> 1. **A New Paradigm, Not Post-Processing**: We are the first to introduce TTA to feedforward 3D reconstruction. This shifts from static inference to dynamic, per-scene model adaptation. Crucially, this is **not post-processing**, which operates on fixed outputs. Instead, we dynamically adapt the model's parameters for each scene during inference, which is fundamentally different from refining a static result.
>
> 2. **Technical novelty**:
> Our technical approach enables this new paradigm with two key contributions:
> - **Self-consistency as new objective.**
> Our objective is straightforward by design, but it is **not suboptimal**. It is based on the universal principle of geometric self-consistency: any partial reconstructions of the same scene must be consistent with one another. For a perfect reconstruction, this self-consistency loss would naturally be zero. This self-consistency loss is never introduced in any paper before.
> - **First use of VPT in 3D**
> While VPT is an established PEFT method, we are the first to apply it to feedforward 3D models under TTA. Adapting visual prompts for dense 3D geometry prediction is non-trivial and requires careful integration.
>
> We will highlight those points in the revision. Thanks!
> > **`>>> W5:`**
> > **There are minor issues, such as the typo "recomstruction" on line 39.**
> ---
> **`>>> A5:`**
>
> We appreciate your reminder and will revise it.

---

> > ### Comment · Reviewer_aSNX · 2025-08-06
> >
> > Thanks for the authors’ efforts in the rebuttal. I am impressed by the significant performance improvements achieved when integrating with VGGT, as well as the remarkably short optimization time of only 14 seconds.
> >
> > That said, I still retain some minor concerns:
> > 1. While fine-tuning model parameters for each scene individually is different from post-processing, I would not consider this a new paradigm. Similar approaches have been explored in the era of generalized NeRFs, such as SparseNeuS. The key difference lies in the fact that per-scene fine-tuning for generalized NeRFs tends to be much slower.
> > 2. I still believe that using self-consistency as the optimization objective may not be the most optimal choice (although it is correct). Because this objective does not fully exploit all available input information. Nevertheless, the authors have demonstrated that this objective works well in practice and delivers impressive results, so this is not a major issue.
> >
> > Overall, I decide to raise my score to borderline accept. And I would recommend the authors to incorporate some of key points mentioned in rebuttal into the the paper.

---

> > > ### Author Response · Authors · 2025-08-06
> > >
> > > Dear Reviewer aSNX,
> > >
> > > Thank you for understanding and for raising the score. We truly appreciate your constructive feedback and insights on our paradigm and training objectives. Your suggestions have been very helpful, and we will incorporate them into the main paper.
> > >
> > > Best regards,
> > >
> > > Authors of Submission 5055

---

### Official Review · Reviewer_dVwe · 2025-07-02

**Clarity:** 2
**Significance:** 2
**Originality:** 2
**Rating:** 4
**Confidence:** 5

**Summary:**

This paper proposes Test3R, a test-time fine-tuning framework aimed at enhancing the performance of feedforward reconstruction methods such as DUSt3R. The authors observe that the output point map of the same image varies when paired with different images. To address this inconsistency, they introduce a prompt tuning strategy applied at test time. Specifically, prompt tuning is implemented by appending and fine-tuning additional tokens. Experimental results demonstrate that the proposed method improves the performance of the base models.

**Questions:**

Most important questions:
- [Same as W1] Provide runtime analysis.
- [Same as W2] How does the used prompt tuning compare to full fine-tuning or LoRA?
- [Same as W4] Directly compare on image pairs with our without test time fine-tuning.


- [Q1] The method is limited to handling three input views. However, in Sec. 5.1, experiments are conducted with 3–5 and 2–4 views. How is this discrepancy addressed?
- [Q2] Can the method be extended to handle a larger number of input views, such as more than 10?
- [Q3] In Fig. 1, the point maps are aligned using only ICP, as DUST3R outputs lack consistent scale. This undermines the reliability of the visualizations. Additionally, no comparable visualization of Test3R is provided.

**Ethical Concerns:**

["NO or VERY MINOR ethics concerns only"]

**Final Justification:**

Thank you for the authors' reply. I will raise my score and recommend that, in the revised manuscript, you revise the “cost-free” claim and clearly state the actual time-consuming nature.

**Limitations:**

yes

**Quality:**

2

**Strengths And Weaknesses:**

**Strengths**:

- Test-time fine-tuning is an effective approach for improving the performance of feedforward reconstruction models.
- The proposed method consistently demonstrates improvements over baseline models.

**Weaknesses**:

- [W1] The authors repeatedly claim that their method is nearly cost-free; however, prompt tuning clearly incurs computational cost. No runtime analysis or comparison is provided.

- [W2] The motivation for using prompt tuning is not sufficiently ablated. While test-scene fine-tuning can naturally improve performance, it remains unclear why prompt tuning is specifically needed. There are lots of other fine-tuning methods that can be used, such as fine-tuning or LoRA.

- [W3] The authors claim the method is “universally applicable and nearly cost-free,” yet all experiments are conducted on DUSt3R/MASt3R/MonST3R, which share the same architecture. It is unclear how well the method generalizes to other models like CUT3R or VGGT, which support more input images.

- [W4] Since the method operates directly on image pairs, qualitative results showing outputs before and after applying Test3R would help demonstrate its effectiveness.

- [W5] Other writing issues:
  - Line 39 typo: "reconstruction."
  - Section 4.1 is redundant and overlaps with content in Lines 31–43.
  - The method operates on only two image pairs, making the ellipsis in the left part of Fig. 2 unnecessary.
  - Line 232 mentions using DUST3R as the backbone, while Line 64 refers to MAST3R and MonST3R. This inconsistency should be clarified.

---

> ### Author Rebuttal · Authors · 2025-07-31
>
> Thank you for your positive feedback and insightful comments! Below, we respond to address your concern. We welcome further discussion to enhance the clarity and effectiveness of our work.
>
> >  **`>>> W1:`**
> >  **The authors repeatedly claim that their method is nearly cost-free; however, prompt tuning clearly incurs computational cost. No runtime analysis or comparison is provided.**
> ---
> **`>>> A1:`** We appreciate your suggestion and apologize for the oversight. The test-time optimization process is relatively short, taking approximately 14 seconds per scene on average with a single GPU. Regarding GPU memory usage during pair-wise prediction, Test3R consumes 2.57 GB, compared to 2.54 GB for vanilla DUSt3R. We consider this additional memory footprint negligible.
>
> >  **`>>> W2:`**
> >  **The motivation for using prompt tuning is not sufficiently ablated. While test-scene fine-tuning can naturally improve performance, it remains unclear why prompt tuning is specifically needed. There are lots of other fine-tuning methods that can be used, such as fine-tuning or LoRA.**
> ---
>
> **`>>> A2:`**
> Thanks for your suggestion. We conducted experiments on the Office-09 scene on the 7Scenes dataset. We perform fine-tuning of the DUSt3R encoder via two methods: full-parameters fine-tuning and LoRA. The quantitative results are shown in the table below and will be included in the main paper. Specifically, by maximizing cross-pair consistency introduced in Sec.4.2, all three test-time training approaches improve reconstruction quality. However, our method that uses prompts significantly outperforms both the full-parameters and LoRa-based approaches.
> |  Method   | Acc&nbsp;$\downarrow$ | Comp&nbsp;$\downarrow$ | NC&nbsp;$\uparrow$ |
> | :-------: | :-------------------: | :--------------------: | :----------------: |
> |   LoRa    |         0.45          |          0.32          |        0.62        |
> | Fine-tune |         0.28          |          0.42          |        0.64        |
> |  DUSt3R   |         0.62          |          0.51          |        0.54        |
> |   Ours    |         0.14          |          0.20          |        0.69        |
>
> >  **`>>> W3:`**
> >  **The authors claim the method is "universally applicable and nearly cost-free" yet all experiments are conducted on DUSt3R/MASt3R/MonST3R, which share the same architecture. It is unclear how well the method generalizes to other models like CUT3R or VGGT, which support more input images.**
> ---
>
> **`>>> A3:`** Thank you for pointing this out. We would like to clarify that our method remains **applicable** to multi-image input methods such as VGGT[1]. Specifically, we insert the prompts into the image tokens of each input image, in a manner similar to VGGT’s handling of the register token. During test-time training, it still takes a triplet as input to predict two pointmaps under the same reference view but paired with different source views as introduced in Sec.4.2. To validate the effectiveness of the model, we conducted an evaluation on the Office-07 scene on the 7Scenes dataset. Following the default setting in Sec.5, the number of triplets as test-time training sets is 125, and we optimize the prompt using only one epoch. The final results are shown in the table below. It indicates that our method significantly enhances both Acc. and Comp. of the pointmap predicted by VGGT[1]. This further provides evidence of the "universally applicable" nature of our method. This experiment will be included in the main paper.
> |  Method   | Acc&nbsp;$\downarrow$ | Comp&nbsp;$\downarrow$ | NC&nbsp;$\uparrow$ |
> | :-------: | :-------------------: | :--------------------: | :----------------: |
> |   VGGT    |         0.27          |          0.20          |        0.71        |
> | VGGT+Ours |         0.11          |          0.12          |        0.71        |
>
> [1] VGGT: Visual Geometry Grounded Transformer.
>
>
> >  **`>>> W4:`**
> >  **Since the method operates directly on image pairs, qualitative results showing outputs before and after applying Test3R would help demonstrate its effectiveness.**
> ---
>
> **`>>> A4:`** We apologize we are currently unable to provide similar visualizations here due to NeurIPS policy restrictions. However, we provide a comparison of depthmap consistency in Fig. 5 in the Appendices. This comparison can also illustrate the improvement in pointmap consistency, according to Eq.9. As shwon in this Figure, our model effectively improves cross-pair consistency, and generates consistent and reliable depth in regions with limited overlap. We will supplement the visualization results of the pointmap in subsequent versions.
>
> >  **`>>> W5:`**
> >  **Other writing issues.**
> ---
> **`>>> A5:`**
>
> Thanks for your suggestions for our writing issues; they are helpful in improving the clarity and readability.
> * **Line 39 typo: "reconstruction."**
>     * We will revise it.
> * **Section 4.1 is redundant and overlaps with content in Lines 31–43.**
>     * We will reorganize these two parts accordingly. An overview will be provided in the Introduction section, followed by a detailed description in Section 4.1.
> * **The method operates on only two image pairs, making the ellipsis in the left part of Fig. 2 unnecessary.**
>     *  We will revise the figure.
> * **Line 232 mentions using DUST3R as the backbone, while Line 64 refers to MAST3R and MonST3R. This inconsistency should be clarified.**
>     * We would like to clarify that the experiments in Sec.5 are all based on the DUSt3R(Line232). We applied Test3R to MASt3R and MonSt3R to demonstrate the universal applicability of our model(Line 64). We apologize for the misunderstanding and will correct the description in Line 232.
>
> >  **`>>> Q1:`**
> >  **The method is limited to handling three input views. However, in Sec. 5.1, experiments are conducted with 3–5 and 2–4 views. How is this discrepancy addressed?**
> ---
>
> **`>>> A1:`** Sorry for any confusion. We would like to clarify that our model can handle an arbitrary number of images. The input during test-time training is three images. Specifically, in Sec.5.1, we consider all input viewpoints to construct the triplets. Subsequently, these triplets serve as the test-time training set to optimize our model.
>
> >  **`>>> Q2:`**
> >  **Can the method be extended to handle a larger number of input views, such as more than 10?**
> ---
>
> **`>>> A2:`** **Yes.** Our model can handle an arbitrary number of images.
>
> >  **`>>> Q3:`**
> >  **In Fig. 1, the point maps are aligned using only ICP, as DUST3R outputs lack consistent scale. This undermines the reliability of the visualizations. Additionally, no comparable visualization of Test3R is provided.**
> ---
>
> **`>>> A3:`** Thanks for pointing this out. In fact, we consider scale inconsistency to be **a specific part of cross-pair inconsistency**. If the model can fully adapt to the scene, the scale of its pointmaps would also be consistent. Therefore, we only use ICP to align the pointmap.
>
> **Visualization.** Due to NeurIPS policy restrictions, similar visualizations of Test3R are not able to be included in the rebuttal phase. We will supplement the visualization results of the pointmap in revision. We would like to kindly invite you to refer to Fig.5 in the appendix, which provides a comparison of cross-pair consistency on the depthmap. It can reflect our improvement for pointmap consistency, according to Eq.9.

---

> > ### Comment · Reviewer_dVwe · 2025-08-04
> > **concerns**
> >
> > Thanks for the authors' rebuttal, but I still have some concerns:
> >
> > - "The test-time optimization process is relatively short, taking approximately 14 seconds per scene on average with a single GPU." How is this time estimated? Factors such as the number of input views and image resolution will affect the fine-tuning time. I believe that requiring such a long fine-tuning time cannot be considered cost-free, especially when integrated with methods such as VGGT, which is fast for multi-view inputs.
> > - For comparison with other fine-tuning methods, only using one scene is biased and not supportive. This is also true for integrating with VGGT.

---

> ### Author Response · Authors · 2025-08-05
>
> I sincerely thank the Reviewer-dVwe for the new suggestions!
>
> >  **`>>> Q1:`**
> > **Test-time Optimization.**
> ---
>
> **`>>> A1:`** Thanks for the question! We estimate the time on the 7Scenes dataset. For the test-time training phase, we process all available image triplets, up to a maximum of 165. If a scene contains more than 165 triplets, we randomly sample this maximum amount. With a batch size of 2, this takes at most $165/2\approx 83$ iterations. Each image is resized to a resolution of $512\times 384$, matching the setting in the DUSt3R. Under this setup, the average processing time across all scenes is 14 seconds.
>
> **Nearly cost-free.** Sorry for the confusion! While 14 seconds may not seem short on its own, it is negligible compared to the overall runtime of pipelines like DUSt3R and MonST3R. For example, processing 50 images from the 7Scenes dataset already takes 320 seconds with DUSt3R alone. In this context, the additional 14 seconds is relatively minimal, "nearly cost-free".
>
> >  **`>>> Q2:`**
> > **Average results across the entire dataset.**
> ---
> **`>>> A2:`** We appreciate your suggestion! In response, we conducted additional experiments on all 18 scenes in the 7Scenes dataset.
>
> **PEFT evaluation.** We report averaged results across all 18 scenes on the 7Scenes dataset. All experimental settings remain consistent with our default setting provided in Sec.A. The average scores of all scenes are presented in the table below. As shown, leveraging our training objective to maximize cross‑pair consistency enables all three fine‑tuning methods to achieve improvements in the final reconstruction results. The LoRA-based method, direct fine-tuning, and our approach resulted in decreases in Acc. of 0.012, 0.009, and 0.04, respectively.
>
> These results support our claim that prompt tuning is a strong choice for test-time training.
>
> |            Method            | Acc&nbsp;$\downarrow$ | Comp&nbsp;$\downarrow$ | NC&nbsp;$\uparrow$ |
> |:---------------------------:|:--------------------:|:-------------------:|:--------------------------:|
> | DUSt3R                     |      0.146                |    0.181                 |          0.736                 |
> | LoRa                     |       0.134       |    0.139                   |               0.730              |
> | Fine-tune                     |        0.137     |      0.139               |                    0.722        |
> | Ours                     |      **0.105**                |    **0.136**                 |          **0.746**             |
>
> **VGGT Implementation.** We further evaluated the results of implementing our method on VGGT across all 18 scenes from the 7Scenes dataset. For detailed implementation, we specifically insert the prompts into the image tokens of each input image, in a manner similar to VGGT’s handling of the register token. During test-time training, it still takes a triplet as input to predict two pointmaps under the same reference view but paired with different source views as introduced in Sec.4.2. To validate the effectiveness of the model, we follow the default experimental settings and construct the triplets as described in Sec.A. The final results are presented in the table below. We outperform vanilla VGGT on both the Acc and Comp metrics, with only the NC metric showing comparatively lower performance. This demonstrates that **our method is also applicable to multi-view input approaches such as VGGT**.
>
> |            Method            | Acc&nbsp;$\downarrow$ | Comp&nbsp;$\downarrow$ | NC&nbsp;$\uparrow$ |
> |:---------------------------:|:--------------------:|:-------------------:|:--------------------------:|
> | VGGT                     |      0.084        |     0.088                 |      **0.785**                        |
> | VGGT+Ours                     |    **0.051**          | **0.066**                    |                  0.760           |

---

> > ### Comment · Reviewer_dVwe · 2025-08-06
> > **cost-free**
> >
> > Thanks for the further clarification. The fine-tuning comparison on more scenes looks good. However, regarding the time issue, I agree that DUSt3R may take a long time to process a large number of input views due to optimization issues. However, how does it compare to VGGT in terms of time? 14 seconds seems to be longer than the original VGGT inference time.

---

> > > ### Author Response · Authors · 2025-08-07
> > >
> > > Thank you so much for this question! It’s a very sharp observation, and we're genuinely grateful for the chance to elaborate on the runtime.
> > >
> > > **Time cost on VGGT**. You're absolutely right: On VGGT, our cost is relatively higher. Through our experiment, VGGT takes 9 seconds for 50 images, while our method takes an extra 28 seconds on top of that. We see this as a clear trade-off: the additional compute yields improvements in geometric accuracy.
> > >
> > > To provide some context, our work began in January and was initially developed on top of DUSt3R. **The 'cost-free' claim originated from our initial focus on DUSt3R, which was our primary baseline**. As VGGT is a **concurrent work***, it was not the focus of our initial efficiency analysis.
> > >
> > > However, we do realize now that this may have been misleading on VGGT. Thank you for pointing that out, and we will update the paper to precisely state the context of our claim and add a thorough discussion of VGGT.
> > >
> > >
> > > **Future Work on Time Optimization**. We fully agree that efficiency is crucial, and we are actively working on this. We have two main strategies to reduce this cost:
> > > 1. **Reduce the TTA iterations**. We plan to investigate enhanced triplet sampling strategies and more efficient optimization algorithms. We hope to achieve the same accuracy gains with significantly fewer iterations.
> > > 1. **Triplet Objective in Training**. A key opportunity is to incorporate our triplet objective as an unsupervised loss directly into the VGGT training pipeline. This could instill stronger geometric consistency from the start, reducing the need for TTA.
> > >
> > > Of course, we do acknowledge the speed trade-off. But given the consistent improvement on VGGT, we feel it's a worthwhile cost for now. That said, we're actively working on making it faster.
> > >
> > > Thank you again for this insightful feedback!
> > >
> > >
> > > > $[*]$ Note that VGGT *was released on arXiv on March 14th after March 1st and is thus regarded as a concurrent work according to NeurIPS policy.*
> > > >
> > > >  Papers appearing online after March 1st, 2025, are generally considered concurrent to NeurIPS submissions. Authors are not expected to compare to those. ---- **NeurIPS 2025 FAQ for Authors**
> > > ---

---

> > > ### Author Response · Authors · 2025-08-09
> > >
> > > Dear Reviewer dVwe,
> > >
> > > We wish to convey our sincere appreciation for your invaluable feedback, which has been of great help to us. As the discussion deadline approaches, we are keenly anticipating any additional comments you may have.
> > >
> > > We are deeply grateful for your commitment to the review process!
> > >
> > > Best regards,
> > >
> > > Authors of Submission 5055

---

> > > > ### Comment · Reviewer_dVwe · 2025-08-09
> > > > **response**
> > > >
> > > > Thank you for the authors' reply. I will raise my score and recommend that, in the revised manuscript, you revise the “cost-free” claim and clearly state the actual time consuming.

---

> > > > > ### Author Response · Authors · 2025-08-09
> > > > > **Thanks for raising score**
> > > > >
> > > > > Dear Reviewer dVwe,
> > > > >
> > > > > Thank you for acknowledging our work and for raising the score. We sincerely appreciate your constructive and detailed feedback, and **will revise our paper accordingly**!
> > > > >
> > > > > Thank you once again for your recognition, which is highly valuable and incredibly encouraging to us!
> > > > >
> > > > > Best regards,
> > > > >
> > > > > Authors of Submission 5055

---

### Official Review · Reviewer_K6dW · 2025-07-03

**Clarity:** 3
**Significance:** 3
**Originality:** 4
**Rating:** 5
**Confidence:** 4

**Summary:**

This paper introduces Test3R, a novel approach for performing test-time training on pairwise 3D reconstruction models, such as DUSt3R and MASt3R. The authors observe that DUSt3R produces inconsistent reference-view reconstructions when paired with different source views, which can negatively affect the final global alignment needed for a coherent 3D reconstruction. To address this issue, Test3R leverages visual prompt tuning, introducing a few learnable continuous tokens at test time. The method optimizes cross-view consistency of the reference-view pointmap as its training objective. Test3R is primarily evaluated on DUSt3R, demonstrating improvements in both multi-view depth estimation and 3D reconstruction. Additionally, the authors show that the approach generalizes to other pairwise architectures, such as MASt3R and MonST3R.

**Questions:**

1. At test time, are the triplets processed sequentially or in batches? If they are processed in batches, what batch size is used?
2. The reviewer is curious whether the proposed method is compatible with using correspondences (e.g., from an off-the-shelf model), where the training objective is to enforce that points from corresponding pixels are close to each other. This extension could potentially make the approach work with multi-view methods as well.

**Ethical Concerns:**

["NO or VERY MINOR ethics concerns only"]

**Final Justification:**

My concerns have been well resolved in the rebuttal process. I device to maintain my current score.

**Limitations:**

The primary limitation of the proposed method is that it is currently only compatible with pairwise approaches. The authors also discuss additional limitations in the appendix.

**Paper Formatting Concerns:**

No.

**Quality:**

3

**Strengths And Weaknesses:**

Strength:
1. The idea of using test-time training through visual prompt tuning for 3D reconstruction is both interesting and novel. It offers an alternative to classical optimization-based methods, such as bundle adjustment, for improving 3D reconstruction at test time.
2. The experiments show that the proposed method improves the baseline, DUSt3R on both multi-view depth estimation and 3D reconstruction. Moreover, the proposed method also generalizes well to other pairwise methods, including MASt3R and MonST3R.

Weaknesses:
1. Incomplete evaluation: The authors only conduct experiments on depth estimation and 3D reconstruction. The reviewer expects additional results on camera pose estimation.
2. Ablation study: It would be helpful to include a comparison with (naive or LoRa-based) test-time finetuning approaches that adapt the entire model or only the encoder, without introducing extra tokens.
3. The proposed method appears to be specifically designed for pairwise methods, whereas current state-of-the-art approaches are predominantly multi-view models, such as VGGT and CUT3R. This focus may limit the broader applicability and impact of the proposed approach.

---

> ### Author Rebuttal · Authors · 2025-07-31
>
> We would like to thank the reviewer for their insightful feedback and interesting observations. We thank the reviewer for the support of our work, and we address the reviewer’s comments below and will include all the feedback in the revised version of the manuscript.
>
> > **`>>> W1:`**
> > **Evaluation on camera pose estimation.**
> ---
>
> **`>>> A1:`** Thanks for your suggestion. We provide the result on the *cave_2* scene on the Sintel dataset, and we promise that further experimental results will be conducted and included in the main paper. As shown in the table below, Test3R significantly improves the performance of DUSt3R on camera pose estimation in dynamic scenes, even though it has never been trained on any dynamic scene dataset.
> |            Method            | $ATE$&nbsp;$\downarrow$ | $RPE_{trans}$&nbsp;$\downarrow$ | $RPE_{rot}$&nbsp;$\downarrow$ |
> |:---------------------------:|:--------------------:|:-------------------:|:--------------------------:|
> | DUSt3R                     |     1.12        |    0.45                  |       41.38                      |
> | Test3R                     |   **0.78**  |   **0.28**            |            **3.78**                |
>
> > **`>>> W2:`**
> > **Ablation study: It would be helpful to include a comparison with (naive or LoRa-based) test-time finetuning approaches that adapt the entire model or only the encoder, without introducing extra tokens.**
> ---
> **`>>> A2:`**
>
> Thanks for your suggestion. We conducted experiments on the Office-09 scene on the 7Scenes dataset. We perform fine-tuning of the DUSt3R encoder via two methods: naive fine-tuning and LoRA. The quantitative results are shown in the table below and will be included in the main paper. Specifically, by maximizing cross-pair consistency introduced in Sec.4.2, all three test-time training approaches improve reconstruction quality. However, our method, which uses prompts, significantly outperforms both the naive and LoRa-based fine-tuning approaches.
>
> |            Method            | Acc&nbsp;$\downarrow$ | Comp&nbsp;$\downarrow$ | NC&nbsp;$\uparrow$ |
> |:---------------------------:|:--------------------:|:-------------------:|:--------------------------:|
> | LoRa                     |     0.45        |     0.32                 |         0.62                    |
> | Fine-tune                     |        0.28     |      0.42               |                    0.64        |
> | DUSt3R                     |      0.62                |    0.51                 |          0.54                 |
> | Ours                     |      **0.14**                |    **0.20**                 |          **0.69**               |
>
> > **`>>> W3:`**
> > **The proposed method appears to be specifically designed for pairwise methods, whereas current state-of-the-art approaches are predominantly multi-view models, such as VGGT and CUT3R. This focus may limit the broader applicability and impact of the proposed approach.**
> ---
> **`>>> A3:`**
>
> Thank you for pointing this out. We would like to clarify that our method remains **applicable** to multi-image input methods such as VGGT[1]. Specifically, we insert the prompts into the image tokens of each input image, in a manner similar to VGGT’s handling of the register token. During test-time training, it still takes a triplet as input to predict two pointmaps under the same reference view but paired with different source views as introduced in Sec.4.2. To validate the effectiveness of the model, we conducted an evaluation on the Office-07 scene on the 7Scenes dataset. Following the default setting in Sec.5, the number of triplets as test-time training sets is 125, and we optimize the prompt using only one epoch. The final results are shown in the table below. It indicates that our method significantly enhances both Acc. and Comp. of the pointmap predicted by VGGT[1]. This further provides evidence of the "universally applicable" nature of our method. This experiment will be included in the main paper.
> |            Method            | Acc&nbsp;$\downarrow$ | Comp&nbsp;$\downarrow$ | NC&nbsp;$\uparrow$ |
> |:---------------------------:|:--------------------:|:-------------------:|:--------------------------:|
> | VGGT                     |     0.27        |     0.20                 |         0.71                    |
> | VGGT+Ours                     |        0.11     |      0.12               |                    0.71        |
>
> [1] VGGT: Visual Geometry Grounded Transformer.
>
> > **`>>> Q1:`**
> > **At test time, are the triplets processed sequentially or in batches? If they are processed in batches, what batch size is used?**
> ---
> **`>>> A1:`** It's in batches. Batch size is set to 1 in our experimental setup.
>
> >  **`>>> Q2:`**
> > **The reviewer is curious whether the proposed method is compatible with using correspondences (e.g., from an off-the-shelf model), where the training objective is to enforce that points from corresponding pixels are close to each other. This extension could potentially make the approach work with multi-view methods as well.**
> ---
>
> **`>>> A2:`** Yes. Our method is compatible with these training objective. This approach provides a strong complement to our cross-pair consistency training objective. Aligning corresponding pixels enables us to fully leverage the pointmaps of source views, as discussed in Sec.D in the Appendices. Moreover, St4RTrack [1] has demonstrated that this method contributes to improving the model’s reconstruction capability. However, this is not the primary focus of this paper. We consider it as a potential direction for future research, and sincerely appreciate your insightful suggestion.
>
> [1] St4RTrack: Simultaneous 4D Reconstruction and Tracking in the World

---

> > ### Comment · Reviewer_K6dW · 2025-08-03
> >
> > I appreciate the authors’ efforts in the rebuttal. The results presented are interesting and promising; however, most are tested on a single scene. Could the authors also provide results averaged across all testing sequences?

---

> > > ### Author Response · Authors · 2025-08-05
> > >
> > > Thanks for the suggestion! As suggested, we provide the experimental results across the full dataset instead of one scene. These results will be appended to the main paper.
> > >
> > > **Camera pose evaluation.** Following the experiment setting on MonST3R[1], we report the results across 14 scenes on the Sintel dataset. The average score of all scenes is reported in the table below. Compared to DUSt3R, Test3R demonstrates a substantial improvement across all evaluation metrics **even without incorporating any dynamic scenes priors**:
> > > * $ATE$ is reduced from 0.306 to 0.171, indicating significantly more accurate global trajectory estimation.
> > >
> > > * $RPE_{trans}$ drops from 0.176 to 0.088, showing enhanced local translation accuracy.
> > >
> > > * $RPE_{rot}$ decreases dramatically from 9.476 to 1.864, highlighting a major improvement in rotational consistency.
> > >
> > > These results clearly show that Test3R significantly improves both global and local pose accuracy compared to DUSt3R.
> > >
> > >
> > > |            Method            | $ATE$&nbsp;$\downarrow$ | $RPE_{trans}$&nbsp;$\downarrow$ | $RPE_{rot}$&nbsp;$\downarrow$ |
> > > |:---------------------------:|:--------------------:|:-------------------:|:--------------------------:|
> > > | DUSt3R                     |    0.306          |   0.176                    |     9.476                         |
> > > | Test3R                     |   **0.171**   |       **0.088**         |          **1.864**                   |
> > >
> > > [1]MonST3R: A Simple Approach for Estimating Geometry in the Presence of Motion
> > >
> > > **PEFT evaluation.** We report averaged results across all 18 scenes on the 7Scenes dataset. All experimental settings remain consistent with our default setting provided in Sec.A. The average scores of all scenes are presented in the table below. As shown, leveraging our training objective to maximize cross‑pair consistency enables all three fine‑tuning methods to achieve improvements in the final reconstruction results. The LoRA-based method, direct fine-tuning, and our approach resulted in decreases in accuracy of 0.012, 0.009, and 0.04, respectively.
> > >
> > > These results support our claim that prompt tuning is a strong choice for test-time training.
> > >
> > >
> > > |            Method            | Acc&nbsp;$\downarrow$ | Comp&nbsp;$\downarrow$ | NC&nbsp;$\uparrow$ |
> > > |:---------------------------:|:--------------------:|:-------------------:|:--------------------------:|
> > > | DUSt3R                     |      0.146                |    0.181                 |          0.736                 |
> > > | LoRa                     |       0.134       |    0.139                   |               0.730              |
> > > | Fine-tune                     |        0.137     |      0.139               |                    0.722        |
> > > | Ours                     |      **0.105**                |    **0.136**                 |          **0.746**             |
> > >
> > >
> > > **VGGT Implementation.** We further evaluated the results of implementing our method on VGGT across all 18 scenes from the 7Scenes dataset. For detailed implementation, we specifically insert the prompts into the image tokens of each input image, in a manner similar to VGGT’s handling of the register token. During test-time training, it still takes a triplet as input to predict two pointmaps under the same reference view but paired with different source views as introduced in Sec.4.2. To validate the effectiveness of the model, we follow the default experimental settings and construct the triplets as described in Sec.A. The final results are presented in the table below. We outperform vanilla VGGT on both the Acc and Comp metrics, with only the NC metric showing comparatively lower performance. This demonstrates that **our method is also applicable to multi-view input approaches such as VGGT**.
> > >
> > > |            Method            | Acc&nbsp;$\downarrow$ | Comp&nbsp;$\downarrow$ | NC&nbsp;$\uparrow$ |
> > > |:---------------------------:|:--------------------:|:-------------------:|:--------------------------:|
> > > | VGGT                     |      0.084        |     0.088                 |      0.785                        |
> > > | VGGT+Ours                     |    0.051          |  0.066                    |                  0.760           |

---

> > > > ### Comment · Reviewer_K6dW · 2025-08-06
> > > >
> > > > Thanks for sharing the new results! Apologies for missing this earlier — how many input images are used during evaluation? Is it still two, as in test-time training, or are more views used?

---

> > > > > ### Author Response · Authors · 2025-08-07
> > > > >
> > > > > Thanks for the question! The evaluation of Test3R is conducted under standard, widely adopted benchmark settings as follows.
> > > > >
> > > > > * **3D Reconstruction.** Following the experiment setting of CUT3R [1], each scene is used for evaluation, with **3 to 5 available views** in the 7Scenes dataset and **2 to 4 views** in the NRGBD dataset.
> > > > >
> > > > > * **Multi-view Depth.** Following the experiment setting of RobustMVD [1], the evaluation is conducted using **10 views per scene** in the DTU dataset and ETH3D dataset.
> > > > >
> > > > > * **Camera Pose.** Following the experiment setting of MonST3R[3], the evaluation involves **50 views**.
> > > > >
> > > > > [1] Continuous 3D Perception Model with Persistent State.
> > > > >
> > > > > [2] A Benchmark and a Baseline for Robust Multi-view Depth Estimation.
> > > > >
> > > > > [3] MonST3R: A Simple Approach for Estimating Geometry in the Presence of Motion.

---

> > > > > > ### Comment · Reviewer_K6dW · 2025-08-09
> > > > > >
> > > > > > Thanks for the reply! My concerns have been fully addressed, and I’ve decided to maintain my current score.

---

### Official Review · Reviewer_MVdH · 2025-07-03

**Clarity:** 3
**Significance:** 3
**Originality:** 3
**Rating:** 4
**Confidence:** 3

**Summary:**

This paper introduces Test3R, a test-time learning technique designed to improve the geometric consistency of pairwise dense matching methods like DUSt3R for 3D reconstruction. The authors identify that methods based on pairwise predictions can produce pointmaps that are imprecise and mutually inconsistent when a reference image is paired with different source views. Test3R addresses this by enforcing geometric consistency at test time. For a given image triplet
(I1​,I2​,I3​), the method generates two reconstructions relative to the common image I1​ (from pairs (I1​,I2​) and (I1​,I3​)) and then optimizes the network via a self-supervised loss to maximize the geometric alignment between them. This optimization is performed efficiently using prompt tuning. Experiments show that this approach significantly improves the performance of DUSt3R on 3D reconstruction and multi-view depth estimation tasks, achieving state-of-the-art results.

**Questions:**

1. Could you provide a more detailed analysis of the test-time computational overhead? For example, for a typical scene in the 7Scenes or DTU datasets, how many optimization iterations are performed, and what is the total processing time compared to a single inference pass of the original DUSt3R? This is critical for understanding the practical viability of the method.

2. How are the image triplets (I1​,I2​,I3​) selected from the test set? Is the process random, or is there a specific strategy involved? How does the performance of Test3R vary with different numbers of triplets or different selection strategies (e.g., prioritizing wider baselines)?

**Ethical Concerns:**

["NO or VERY MINOR ethics concerns only"]

**Final Justification:**

The author's rebuttal has solved most of my concerns. Hence, I decided to keep my positive rating.

**Limitations:**

yes

**Paper Formatting Concerns:**

I did not notice any major formatting issues.

**Quality:**

3

**Strengths And Weaknesses:**

Strengths:
1. The core idea of Test3R is simple yet effective. Enforcing consistency between two different pairwise predictions that share a common view is an effective self-supervision signal that directly targets the problem with pairwise reconstruction methods.
2.  The method demonstrates significant performance improvements over its baseline (DUSt3R) and achieves state-of-the-art or competitive results on multiple challenging benchmarks for both 3D reconstruction and multi-view depth estimation.
3. The use of visual prompt tuning for the test-time optimization makes sense. It allows the model to adapt to new scenes while keeping the pretrained backbone frozen, making the process far more efficient than fine-tuning the entire network. The paper also demonstrates good generalization by applying the technique to other models like MAST3R and MonST3R.

Weaknesses:
1. The main concern is the practical cost of performing optimization at inference time. While the paper claims the method is "nearly cost-free", it lacks a detailed analysis of the actual test-time overhead. For any given scene, the model must run multiple forward/backward passes to optimize the prompts. This could be a significant drawback for applications requiring fast inference, and the paper does not quantify this trade-off (e.g., time per scene vs. baseline).
2. The method relies on sampling image triplets from the test set, but the strategy for this selection is not discussed. The performance could be sensitive to which triplets are chosen (e.g., wide vs. narrow baseline pairs), how many triplets are used for optimization, and how they are sampled. This underspecified component makes the method less reproducible and its robustness unclear.

---

> ### Author Rebuttal · Authors · 2025-07-31
>
> We thank the reviewer for the constructive comments. We provide our feedback as follows.
>
> > **`>>> W1:`**
> > **Practical cost of performing optimization at inference time.**
> ---
>
> **`>>> A1:`**  Our maximum test‑time training duration is approximately 14 seconds on a single GPU. This remains consistent even with a large number of views.
>
> **Why nearly cost-free?** Test3R introduces only 14 seconds of extra time overhead, which we consider negligible during the entire process. In fact, the majority of Test3R’s time consumption comes from the pair-wise prediction and global alignment introduced by DUSt3R. This proportion increases as the number of images grows. For example, for a set of 50 images on the 7Scenes dataset, pair-wise prediction takes nearly 250s, and global alignment consumes nearly 70 seconds.
>
> > **`>>> W2:`**
> > **Strategy for triplet selection. Discussion on the performance and which triplets are chosen, how many triplets are used for optimization, and how they are sampled.**
> ---
>
> **`>>> A2:`**
> * **Triplet sampling strategy.** We describe our triplets sampling strategy in Sec.A in the Appendices, and we will relocate this content to the experiment section(Sec.5). Considering a scene with $n$ images, the total number of triplets is $n^3$ for test-time training. For computational efficiency, if the number of triplets exceeds 165, we randomly sample 165 triplets as the test-time training set for each epoch.
>
> Here, we discuss the number of triplets and the sampling strategy on the Office-09 scene on the 7Scenes dataset. This scene consists of five sequential images; therefore, 125 triplets are used in each epoch.
>
> * **The number of triplets is used.** We report the results under different epoch settings to evaluate the effect of the number of triplets. As shown in the table below, Test3R significantly enhances the reconstruction quality of the scene just using one epoch for training, which is our default setting. With more epochs, the optimization leverages an increasing number of triplets, resulting in an improvement in reconstruction quality.
>
> |            Epoch            | Acc&nbsp;$\downarrow$ | Comp&nbsp;$\downarrow$ | NC&nbsp;$\uparrow$ |
> |:---------------------------:|:--------------------:|:-------------------:|:--------------------------:|
> | Vanilla DUSt3R                     |     0.62        |     0.51                 |         0.54                    |
> | 1                     |     0.14        |     0.20                 |         0.69                    |
> | 2                    |        0.06     |      0.08               |                    0.70        |
> | 3                     |      0.04                |    0.04                 |          0.72                 |
>
> * **Triplets sampling**. Given that this scene is captured sequentially, short-baseline triplets are generated by increasing the frequency of source views close to the reference view. Conversely, this results in the wide-baseline experiment. As shown in the table below, the reconstruction results of these three methods are comparable. This demonstrates the robustness of our method, as it is largely insensitive to the selection of triplets.
>
> |            Epoch            | Acc&nbsp;$\downarrow$ | Comp&nbsp;$\downarrow$ | NC&nbsp;$\uparrow$ |
> |:---------------------------:|:--------------------:|:-------------------:|:--------------------------:|
> | Vanilla DUSt3R                     |     0.62        |     0.51                 |         0.54                    |
> | Ours                     |     0.14        |     0.20                 |         0.69                    |
> | Wide-Baseline                     |  0.13            |  0.18                    |     0.71                        |
> | Short-Baseline                     |    0.15                  |       0.21              |     0.69                       |
>
>
> > **`>>> Q1:`**
> > **A detailed analysis of the test-time computational overhead. How many optimization iterations are performed? What is the total processing time compared to the original DUSt3R?**
> ---
> **`>>> A1:`** We report the time consumption of Test3R on the Office scene with a total of 50 images. We follow our default setting and use 165 triplets to optimize Test3R for only one epoch. Therefore, there are 165 optimization iterations performed. In terms of time consumption, this training process only takes 14 seconds. After that, the pair-wise prediction and global alignment introduced by vanilla DUSt3R are performed using 320 seconds.
>
> > **`>>> Q2:`**
> > **How are the image triplets (I1,I2,I3) selected?  How does the performance of Test3R vary with different numbers of triplets or different selection strategies?**
> ---
> **`>>> A2:`** Sorry for any confusion.
>
> * **Triplet sampling strategy.** We describe our triplets sampling strategy in Sec.A in the Appendices, and we will relocate this content to the experiment section(Sec.5). Considering a scene with $n$ images, the total number of triplets is $n^3$ for test-time training. For computational efficiency, if the number of triplets exceeds 165, we randomly sample 165 triplets as the test-time training set for each epoch.
> * **Performance variation.** As we mentioned in W2, the reconstruction quality improves as the number of triplets increases, and does not exhibit any significant differences when different baselines are selected.

---

> > ### Comment · Reviewer_MVdH · 2025-08-04
> >
> > Thank you for the detailed and constructive rebuttal. I appreciate that you have provided new experiments and clarifications that effectively address my main questions regarding the test-time computational overhead and the triplet selection strategy.
> > At present, I am leaning towards maintaining my initial rating.

---

> > > ### Author Response · Authors · 2025-08-05
> > >
> > > Dear Reviewer MVdH,
> > >
> > > Thank you very much for your comments and support! We will carefully incorporate your suggestions to improve the paper.
> > >
> > > If you have any further questions or additional points you’d like us to address, please don’t hesitate to let us know.
> > >
> > > Best regards,
> > >
> > > Authors of Submission 5055

---

### Note · Authors · 2025-08-14

Dear Program Chair, Senior Area Chair, Area Chair, and Reviewers,

We thank all reviewers for the constructive and positive comments:
* Simple yet effective(Reviewer MVdH, dVwe).
* Interesting and novel(Reviewer K6dW)
* Performance improvements(Reviewer MVdH, K6dW, dVwe, aSNX).
* Clearly written(Reviewer aSNX)

We truly appreciate that **reviewers mentioned we have addressed their concerns and raised questions**. Below, we summarize the main points that the reviewers were most concerned about:

* **Design choice of prompt tuning.** As a response, we applied our training objective to three fine-tuning methods for validation: direct fine-tuning, LoRa-based tuning, and our prompt tuning. Their performance was evaluated on the 7scenes dataset. Leveraging our training objective improves the final reconstruction results across all three fine-tuning methods. Notably, **our prompt tuning-based method surpasses both direct fine-tuning and LoRa-based approaches**.

* **Universal applicability.** To support our claim, our method was applied to two categories of models: pairwise methods like MonST3R and MAST3R, and multi-view models like VGGT. Their performance was evaluated on the 7scenes dataset. The experimental results demonstrate that **applying our method to these models significantly enhances the final reconstruction quality**. These results support our claim that our method is universally applicable to 3D geometry foundation models.

* **Time consumption.** For clarification, we report the processing time for test-time training and the overall runtime of pipelines like DUSt3R. While the processing time may not seem short on its own, **it is negligible compared to the overall runtime of pipelines** like DUSt3R and MonST3R.

Finally, we are deeply grateful that reviewers have recognized our improvement and potential to inspire and advance future exploration in 3D foundation models.

Best regards,

Authors of Submission 5055

---

### Decision · Program_Chairs · 2025-09-17

**Decision:**

Accept (poster)

**Comment:**

The paper received 4 reviews. The authors-reviewers discussion clarified most points raised in the initial reviews. A consensus that the paper brings a contribution to 3D reconstruction and brings a method with good performance was reached (in both reconstruction quality and computation time), and that the paper can be revised to meet the requirements for publication based on the authors-reviewers discussion. The scores were revised accordingly. The paper could be accepted as poster.